# Comprehensive occupational health services for healthcare workers in Zimbabwe during the SARS-CoV-2 pandemic

Fungai Kavenga[1], Hannah M. Rickman[2]*, Rudo Chingono[1], Tinotenda Taruvinga[1,3], Takudzwa Marembo[4], Justen Manasa[4], Edson Marambire[1], Grace McHugh[1], Celia L. Gregson[5], Tsitsi Bandason[1], Nicol Redzo[1], Aspect Maunganidze[6], Tsitsi Magure[7], Chiratidzo Ndhlovu[8], Hilda Mujuru[9], Simbarashe Rusakaniko[10], Portia Manangazira[11], Rashida A. Ferrand[1,2], Katharina Kranzer[1,2,11,12]

1 Biomedical Research and Training Institute, Harare, Zimbabwe, 2 Clinical Research Department, London School of Hygiene & Tropical Medicine, London, United Kingdom, 3 Department of Global Health and Development, London School of Hygiene & Tropical Medicine, London, United Kingdom, 4 African Institute of Biomedical Science and Technologies Laboratory, Harare, Zimbabwe, 5 Musculoskeletal Research Unit, Translational Health Sciences, Bristol Medical School, University of Bristol, Bristol, United Kingdom, 6 Department of Surgery, College of Health Sciences, University of Zimbabwe, Harare, Zimbabwe, 7 Department of Obstetrics and Gynaecology, College of Health Science, University of Zimbabwe, Harare, Zimbabwe, 8 Department of Medicine, University of Zimbabwe College of Health Sciences, Harare, Zimbabwe, 9 Department of Paediatrics and Child Health, University of Zimbabwe College of Health Sciences, Harare, Zimbabwe, 10 Department of Community Medicine, College of Health Sciences, University of Zimbabwe, Harare, Zimbabwe, 11 Department of Epidemiology and Disease Control, Ministry of Health and Child Care, Harare, Zimbabwe, 12 Department of Infectious Diseases & Tropical Medicine, Ludwig Maximilian University of Munich, Munich, Germany

☯ These authors contributed equally to this work.
* hannah.rickman@lshtm.ac.uk

**Data Availability Statement:** The minimal data set is available within the paper and its Supporting information files.

## Abstract

### Background

Healthcare workers are disproportionately affected by COVID-19. In low- and middle-income countries, they may be particularly impacted by underfunded health systems, lack of personal protective equipment, challenging working conditions and barriers in accessing personal healthcare.

### Methods

In this cross-sectional study, occupational health screening was implemented at the largest public sector medical centre in Harare, Zimbabwe, during the "first wave" of the country's COVID-19 epidemic. Clients were voluntarily screened for symptoms of COVID-19, and if present, offered a SARS-CoV-2 nucleic acid detection assay. In addition, measurement of height, weight, blood pressure and HbA1c, HIV and TB testing, and mental health screening using the Shona Symptom Questionnaire (SSQ-14) were offered. An interviewer-administered questionnaire ascertained client knowledge and experiences related to COVID-19.

**Funding:** This work was supported by Global Public Health strand of the Elizabeth Blackwell Institute for Health Research, funded under the University of Bristol's QR GCRF strategy (ref: H100004-148. Awarded to CLG, KK, RAF, PM. http://www.bristol.ac.uk/blackwell/health-research/covid-19-research/). This study was additionally funded by UK aid from the UK government (FCDO) (ref 668 303) (KK) https://www.ukaiddirect.org/; and by funding from the government of Canada https://www.international.gc.ca/world-monde/funding-financement/index.aspx The views expressed do not necessarily reflect the policies of the respective governments. RF is funded by a Wellcome Trust Senior Fellowship 206316_Z_17_Z https://wellcome.org/grant-funding The funders had no role in study design, data collection and analysis, decision to publish, or preparation of the manuscript. Hannah Rickman is supported by a Wellcome Trust Clinical PhD Fellowship (Grant number 203905/Z/16/Z).

**Competing interests:** The authors have declared that no competing interests exist.

## Results

Between 27th July and 30th October 2020, 951 healthcare workers accessed the service; 210 (22%) were tested for SARS-CoV-2, of whom 12 (5.7%) tested positive. Clients reported high levels of concern about COVID-19 which declined with time, and faced barriers including lack of resources for infection prevention and control.

There was a high prevalence of largely undiagnosed non-communicable disease: 61% were overweight or obese, 34% had a blood pressure of 140/90mmHg or above, 10% had an HbA1c diagnostic of diabetes, and 7% had an SSQ-14 score consistent with a common mental disorder. Overall 8% were HIV-positive, with 97% previously diagnosed and on treatment.

## Conclusions

Cases of SARS-CoV-2 in healthcare workers mirrored the national epidemic curve. Implementation of comprehensive occupational health services during a pandemic was feasible, and uptake was high. Other comorbidities were highly prevalent, which may be risk factors for severe COVID-19 but are also important independent causes of morbidity and mortality. Healthcare workers are critical to combatting COVID-19; it is essential to support their physical and psychological wellbeing during the pandemic and beyond.

## Background

Healthcare workers are critical in the fight against COVID-19. Multiple studies have shown they are at high risk of SARS-CoV-2 infection [1–3], and tragically thousands of healthcare workers have died of COVID-19 globally [4]. Healthcare workers may also experience significant psychological consequences from the additional workload, challenging conditions within and outside the workplace, and the increased personal risk of acquiring COVID-19 associated with their professional responsibilities [5,6]. Protecting healthcare workers is critical to the COVID-19 pandemic response [7].

In Zimbabwe, the first SARS-CoV-2 case was reported on 20 March 2020, prompting the government to impose a national lockdown and other public health measures such as social distancing [8]. The number of confirmed cases in Zimbabwe remained relatively low until July, when there was a significant increase in the number of people testing SARS-CoV-2 positive and a corresponding rise in the number of individuals presenting to health facilities with COVID-19. By the end of October 2020, a total of 8367 individuals had tested positive for SARS-CoV-2 and 243 COVID-19-related deaths had been reported [9].

Zimbabwe's COVID-19 epidemic unfolded in the context of an overstretched and underfunded healthcare system [10]. The pandemic came immediately after a year-long industrial action by doctors and nurses. Following the country's first COVID-19 death in March 2020, doctors in governmental hospitals took further industrial action over a lack of personal protective equipment (PPE), difficult working conditions and poor remuneration. Three months later nurses joined the work stoppage. Supplies of PPE were limited, and many healthcare facilities lacked safe running water to enable implementation of effective infection prevention and control (IPC) measures [11].

Healthcare workers face threats to their own health other than SARS-CoV-2. Tuberculosis (TB), another air-borne disease, is highly prevalent in Zimbabwe [12], with healthcare workers

at elevated risk compared to the general population [13]. Zimbabweans have a high burden of chronic disease, much of which is undiagnosed, which likely affects healthcare workers to a similar extent to the rest of the population [14]. Such diseases include HIV, with an adult prevalence estimated at 12.7% [15], and non-communicable diseases such as hypertension and diabetes [16–18]. These conditions have the potential to cause significant morbidity and mortality, and are of acute relevance as medical comorbidities have been associated with a high risk of developing severe COVID-19 [19–21].

We implemented comprehensive occupational health services for healthcare workers, including SARS-CoV-2 testing with rapid feedback of results to reduce nosocomial spread, alongside screening for common causes of morbidity and mortality. We further investigated healthcare workers' knowledge and experience of the COVID-19 pandemic and the barriers they faced, to inform appropriate support measures.

## Methods

### Study setting and population

The study was conducted between 27 July and 30 October 2020 at Parirenyatwa Group of Hospitals (PGH) in Harare, Zimbabwe. PGH is the largest public medical centre in Zimbabwe with an inpatient capacity of 1200 beds, general medical and surgical wards, an outpatient wing including an HIV clinic, an eye centre, a mental health and women's hospital, medical and nursing schools and staff residences on site. Normally, over 2000 staff are employed directly by PGH, with many additional domestic and security workers employed by private companies. However, as a result of the national lockdown, the facility moved to "emergency operations", with outpatient services closed from the end of March 2020. Provision of other health services was limited by the industrial action by doctors and nurses. On 9 June 2020, a dedicated wing of the hospital was refurbished and opened as a COVID-19 treatment centre. More than 90 staff were involved in supportive roles such as domestics, security and housekeeping, or in direct care (nurses and nursing assistants) for patients testing positive for SARS-CoV-2. At the "first peak" of the SARS-CoV-2 epidemic in Zimbabwe (July and August 2020) the COVID-19 treatment centre treated up to 30 patients each day.

Study participants were adult clients using the occupational health service who agreed to participate in the research. All were healthcare workers and students, with a deliberately broad scope which included medical and nursing staff but also those working in domestic, administrative, security and other ancillary services. Staff working at any health facility in Harare were free to access the occupational health service. However, given that the service was set-up on the PGH campus and travel restrictions remained in place, most clients were either working at PGH, studied and/or lived on-site.

### Interventions and procedures

With the support of the hospital administration, an occupational health check was offered free of charge during weekdays on an appointment basis. The service was advertised to healthcare workers through fliers and posters, via departmental heads, on work social media platforms and through word-of-mouth. Enrolment was voluntary. All services were offered in an outside space using tents to ensure good ventilation, with social distancing observed. Clients accessing the service were provided with an information sheet and verbal informed consent was obtained and recorded electronically as part of the questionnaire. Clients could opt out of any of the screening tests offered.

The occupational health screening included measurement of height, weight, temperature, oxygen saturation and blood pressure, and point-of-care HbA1c testing (SD Biosensor,

Singapore). Definitions for raised blood pressure, HbA1c and body mass index (BMI) followed World Health Organization guidelines [22,23]. Systolic blood pressure of 140mmHg or above, and/or diastolic blood pressure of 90mmHg or above was considered elevated, and systolic blood pressure of 180mmHg or above, and/or diastolic blood pressure of 120mmHg or above, as severely elevated. An HbA1c of 6.5% or above was considered diagnostic of diabetes. An individual with a BMI of 25 to less than 30 kg/m$^2$ was considered overweight, and of 30 or above obese.

In addition, clients were offered HIV testing, either as a provider-delivered rapid blood test (Alere Determine HIV 1/2, USA) or an oral mucosal self-test (OraSure Technologies, USA), which could be self-administered onsite or taken offsite. Those with positive HIV tests were referred for confirmatory testing according to national guidelines. Clients were screened using the Shona Symptom Questionnaire (SSQ-14) that was developed and validated in Zimbabwe for symptoms of mental illness, with a score of 8 or above considered suggestive of common mental disorder [24]. Participants were asked to self-complete the SSQ questionnaire, and then submitted this to a research nurse who checked the score and looked for "red flag" responses (suicidal ideas and/or hallucinations) which prompted referral to mental health services regardless of the SSQ-14 score.

All clients were asked about symptoms of COVID-19 including fever, cough, coryzal symptoms, joint pain, headache and loss of smell or taste. Clients with one or more COVID-19 symptom, or a measured temperature of >37.5˚C were offered a nasopharyngeal swab for SARS-CoV-2 polymerase chain reaction (PCR) testing. A WHO TB symptom screen [25] was performed, and if positive clients were asked to provide a spontaneous sputum sample for Xpert MTB/RIF testing (Cepheid, USA).

SARS-CoV-2 test results were returned to clients within 48 hours. Negative SARS-CoV-2 tests were communicated either by phone or SMS/Whatsapp. Clients with positive results were contacted by telephone and given advice on IPC measures for themselves and household contacts. Severity of symptoms was assessed and referral made for admission if warranted. The IPC department leads were informed of any positive cases so that they could offer extra support to their staff.

All screening services were provided free of charge. Clients screening positive for diabetes, hypertension or mental illness, and those newly diagnosed with HIV or TB, were referred to appropriate services (PGH staff clinic, or other public or private facilities, according to client preference) for ongoing management. Costs for follow-up varied depending upon where the client chose to access services and whether or not they had medical insurance. Most public services in Zimbabwe charge an access fee, which is less expensive than the costs of private providers (for example, a consultation in a public sector primary healthcare clinic costs USD$4). Clients with a positive SARS-CoV-2 test and those with a high SSQ-14 score were offered referral to the counselling service unit; those who accepted were telephoned by a counsellor so no cost was incurred by the client.

## Laboratory methods

Clients consenting to SARS-CoV-2 testing had a nasopharyngeal swab collected, which was immediately placed in viral transport medium (VTM) and transported to the laboratory for testing within six hours of collection. At the laboratory RNA was extracted from 200 ul of the VTM using NUCLISENS easyMAG (bioMérieux, USA) as per manufacturer's instructions. Multiplex SARS-CoV-2 real time reverse transcriptase PCR (rtPCR) was performed using Taq-Path™ COVID-19 kit (Thermo Fisher, USA). The assay detects three viral genes, ORF 1ab, N gene and S as well as the bacteriophage MS2 process control. 10 ul of RNA was used in 30 ul

reaction mixes, which were amplified and analysed on AB7500 (Applied Biosystems, USA) rtPCR machine. The rtPCR conditions were as follows: incubation of 2 minutes at 25˚C, reverse transcriptase at 53˚C for 10 minutes, activation at 95˚C for 2 minutes and 40 cycles of denaturation at 95˚C for 3 seconds and annealing/extension at 60˚C for 30 seconds.

### Data collection and statistical analysis

While the clients waited for screening, a trained researcher administered a baseline questionnaire, which included questions on past medical history, knowledge about COVID-19 prevention and management, attitudes and behaviours related to COVID-19 and availability of PPE and other resources in their workplace.

Data were collected using SurveyCTO, a secure electronic mobile data collection and management system loaded onto tablets. Uploaded data were extracted from the SurveyCTO server and saved to a Microsoft SQL Server hosted at the Biomedical Research and Training Institute. Data analysis was performed using Stata version 13 (College Station, USA).

There was minimal missing data, but denominators are specified when relevant. Changes over time in levels of concern and feeling about COVID-19 were inspected using Locally Weighted Scatterplot Smoothing (lowess) plots, with simple linear and logistic regression then used to explore the association. For BMI, blood pressure, HbA1c and SSQ-14 outcomes, the accepted clinically-relevant cut-offs were used to dichotomise outcomes. Univariable and multivariable logistic regression were performed to assess associations of covariates with the outcomes. Lowess plots were used to inspect the relationship of the outcomes with age, and spline terms introduced where there was evidence of a non-linear relationship. Age and sex were considered possible covariates for all conditions and were included in multivariable models. For raised blood pressure and HbA1c, raised BMI was also considered as a possible independent variable and included in multivariable analyses; for raised SSQ-14, association with self-reported level of concern about COVID-19 was also investigated. Variance inflation factors were used to exclude significant collinearity in covariables. Prevalence and 95% confidence intervals were calculated for clinical conditions, with the numerator including clients who had a pre-existing diagnosis of the condition but may not necessarily have had an abnormal test in occupational health screening.

COVID-19 case numbers by date for Zimbabwe were extracted from the Worldometer website [9].

### Ethics

Ethical approval was granted by the Institutional Review Board of the Biomedical Research and Training Institute, the Medical Research Council of Zimbabwe (MRCZ/A/2627) and the London School of Hygiene and Tropical Medicine ethics committee (22514). Clients were given an information sheet about the services offered. The study requested and was granted a waiver allowing verbal rather than written consent to be obtained. Verbal consent was recorded by trained researchers using the SurveyCTO platform.

## Results

### 1. Client characteristics and comorbidities

In the 14-week period between 27 July and 30 October 2020, 951 healthcare workers accessed the service and were enrolled in the study. The median age was 35 (interquartile range [IQR] 29–42) years and 784 (82%) were female (Table 1). A wide range of healthcare professions was included; the most represented were nursing or midwifery (n = 404, 42%), and domestic

**Table 1. Characteristics of 951 enrolled healthcare workers.**

| Age, years | Median | IQR |
|---|---|---|
| | 35 | (29–42) |
| **Sex** | N | % |
| Male | 167 | (17.6%) |
| Female | 784 | (82.4%) |
| **Role** | N | % |
| *Clinical roles* | *545* | *(57.3%)* |
| Nursing and midwifery | 404 | (42.5%) |
| Allied health professionals | 65 | (6.8%) |
| Medical | 45 | (4.7%) |
| Dental | 31 | (3.3%) |
| *Non-clinical roles* | *406* | *(42.7%)* |
| Cleaning/domestic services | 203 | (21.3%) |
| Security/maintenance/porters/drivers | 104 | (10.9%) |
| Administrative | 71 | (7.5%) |
| Laboratory | 15 | (1.6%) |
| Other | 13 | (1.4%) |
| **Highest level of education** | N | % |
| Did not complete primary | 2 | (0.2%) |
| Primary | 6 | (0.6%) |
| Secondary O-Level | 407 | (42.8%) |
| Secondary A-Level | 114 | (12.0%) |
| Diploma after secondary | 280 | (29.4%) |
| University | 142 | (14.9%) |
| **Medical insurance** | N | % |
| Medical insurance | 598 | (62.9%) |
| No medical insurance | 353 | (37.1%) |
| **Medical history**[1] | N | % |
| No medical conditions | 733 | (77.1%) |
| Hypertension | 122 | (12.8%) |
| HIV | 70 | (7.4%) |
| Diabetes | 19 | (2.0%) |
| Asthma/COPD | 17 | (1.8%) |
| Gastritis/peptic ulcer disease | 5 | (0.5%) |
| Cardiovascular disease | 3 | (0.3%) |
| Epilepsy | 2 | (0.2%) |
| Other[2] | 4 | (0.4%) |

IQR: Interquartile range. COPD: Chronic Obstructive Pulmonary Disease.

[1]No participants reported a history of: renal disease, previous tuberculosis, malignancy or psychiatric problems.

[2]Anaemia (1), hayfever (1), hearing loss (1), migraines (1).

services (n = 203, 21%). In total, 733 (77%) clients reported no known previous medical conditions; the most commonly reported diagnoses were hypertension (n = 122, 13%) and HIV (n = 70, 7%). Overall, 902 (95%) reported they had had a previous HIV test and all of those with known HIV reported taking anti-retroviral therapy. About two-thirds of clients (598, 63%) had some form of medical insurance.

## 2. COVID-19 contact, symptoms and test results

Overall, 122 (13%) clients reported having had contact with someone with COVID-19; 70 (57%) reported that they were wearing appropriate PPE at the time. Almost a quarter of clients (n = 213, 22%) reported at least one symptom of: cough, fever, night sweats, loss of taste, loss of smell, fatigue, sneeze, runny nose, headache, joint pains, sore throat, diarrhoea or mouth ulcers, or a recorded fever >37.5˚C. Of the 213 clients with symptoms or a raised temperature, 187 (88%) were tested for SARS-CoV-2. A further 23 clients without symptoms were also tested, of whom 20 (87%) reported contact with someone with COVID-19. Overall, 210 SARS-CoV-2 swabs were performed on healthcare workers, of which 12 (5.7%) were positive.

In the first three weeks of the study, 52/66 (79%) of those presenting to services reported COVID-19 symptoms. Six clients reporting symptoms did not want to be tested, and therefore 46 symptomatic clients and an additional five who did not report symptoms (total 51), were tested for SARS-CoV-2 (S1 Table). Six (12% of those tested; 9% of those enrolled) had a positive result (Fig 1).

The proportion of clients reporting symptoms suggestive of SARS-CoV-2 declined over time, from 79% to 71/251 (28%) and 87/309 (14%) in weeks 1–3, 4–6 and 7–14 respectively. In week 4–6, 79/251 (31%) clients were tested for SARS-CoV-2 (66 symptomatic and 13 asymptomatic), and six had a positive SARS-CoV-2 test (2% of those enrolled and 8% of those

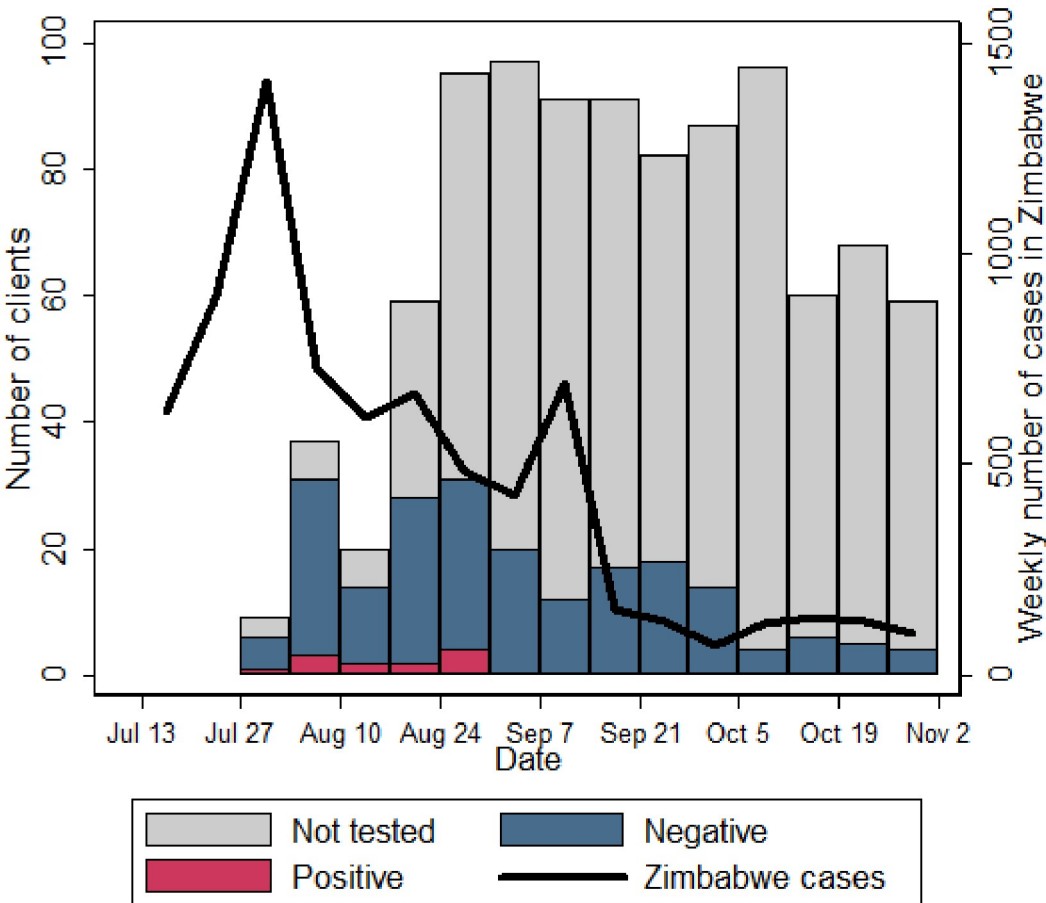

**Fig 1. Numbers of healthcare workers (clients) enrolled and tested for SARS-CoV-2, overlain with weekly national case reports from Zimbabwe *(all dates—2020).***

tested). There were no further positive test results between 31 August and 30 October 2020. All clients testing positive for SARS-CoV-2 were successfully contacted by telephone; at the time of review, four (33%) were asymptomatic and the remaining eight had only mild symptoms. All made an uncomplicated recovery and none required hospital admission. However, 5 (42%) requested referral to counselling services.

## 3. COVID-19 safety, knowledge, attitudes and behaviour

When asked about availability of resources in their workplace, less than half (n = 435, 46%) stated running water was "always" available, and 753 (79%) stated that soap was "always" available (Table 2) In total, 884 (93%) clients reported they had been given a mask to wear, but of these, 431 (49%) stated there were times when no mask was available. Most (n = 742, 78%) reported having received training on SARS-CoV-2 IPC. When asked about the factors which prevented them from effectively protecting themselves from SARS-CoV-2 in the workplace

**Table 2. Healthcare workers' responses to questions about COVID-19 knowledge, attitudes, resources and challenges (n = 950).**

| Are these resources *always* available at your workplace? | |
| --- | --- |
| Water | 435 (45.8%) |
| Mask | 453 (47.7%) |
| Bleach | 688 (72.4%) |
| Soap | 753 (79.3%) |
| Hand sanitizer | 847 (89.2%) |
| **What barriers prevent you from protecting yourself against COVID-19?** | |
| Loss or decrease of wages | 655 (68.9%) |
| Increasing prices in the market | 604 (63.6%) |
| Lack of reliable information about COVID-19 | 523 (55.1%) |
| Shortage of masks | 489 (51.5%) |
| Shortage of water | 474 (49.9%) |
| Shortage of gloves | 397 (41.8%) |
| Shortage of hand sanitizer | 314 (33.1%) |
| Shortage of soap | 191 (20.1%) |
| **Knowledge questions (number and % answering correctly)** | |
| There is no effective cure for COVID-19 *(true)* | 763 (80.3%) |
| Not all people with SARS CoV-2 develop severe disease *(true)* | 755 (79.5%) |
| SARS-CoV-2 cannot be transmitted when symptoms are not present *(false)* | 677 (71.3%) |
| Severe disease is more likely in elderly, chronic illness and obesity *(true)* | 888 (93.5%) |
| Isolation of infected people effectively reduces spread *(true)* | 919 (96.7%) |
| Contacts of COVID-19 should quarantine for 14 days *(true)* | 885 (93.2%) |
| Handwashing with soap is effective against SARS-CoV-2 *(true)* | 894 (94.1%) |
| 0.5% bleach can be used to decontaminate surfaces of SARS-CoV-2 *(true)* | 883 (92.9%) |
| 50% ethanol can be used to decontaminate surfaces of SARS-CoV-2 *(false)* | 266 (28.0%) |
| 70% ethanol can be used to decontaminate surfaces of SARS-CoV-2 *(true)* | 653 (68.7%) |
| Boiling water can be used to decontaminate surfaces of SARS-CoV-2 *(false)* | 388 (40.8%) |
| *Median (interquartile range) knowledge score (out of maximum 11)* | 8 (8–9) |
| **How do you feel about the coronavirus pandemic?** | |
| Very fearful | 179 (18.8%) |
| Fearful | 155 (16.3%) |
| Fearful, but optimistic | 428 (45.1%) |
| Neutral | 188 (19.8%) |

and community, 474 (50%) cited shortage of water, and 489 (51%) shortage of masks. Other factors mentioned included loss or decrease of wages (n = 655, 69%), increasing prices in the market (n = 604, 64%) and lack of reliable information about COVID-19 (n = 523, 55%).

Clients were asked how serious they felt the COVID-19 situation was, on a scale of 1 (not serious) to 10 (very serious). The median score was 6 (IQR 4–8) with a significant decline over time, from 9 (IQR 8–10) in the first week of enrolment, to 4 (IQR 3–5) by the fourteenth week (Fig 2) (p<0.001). When asked how they felt about COVID-19, the most common response (n = 428, 45%) was "fearful, but optimistic"; 334 (35%) answered "fearful" or "very fearful" and the remainder (n = 188, 20%) were "neutral"; the latter proportion increased over time, from 9% in the first three weeks, to 28% in the last four weeks (p<0.001).

Clients generally had good knowledge about SARS-CoV-2, answering a median 9 (IQR 8–9) questions correctly on an 11-point knowledge questionnaire (Table 2). In total, 919 (97%) answered that isolation of infected people effectively reduces spread of SARS-CoV-2, and 885 (93%) that contacts of those with SARS-CoV-2 should quarantine for 14 days. However, only 677 (71%) answered that SARS-CoV-2 could be transmitted by people who did not have symptoms, and 755 (80%) that not everyone with SARS-CoV-2 develops severe disease.

## 4. Other medical conditions

The median BMI was 26.3 kg/m$^2$ (IQR 23.0–30.8), with 319 (34%, 95% confidence interval [CI] 31–37%) of clients overweight and a further 263 (28%, 95% CI 25–31%) obese (Table 3). BMI increased by a mean of 0.20 kg/m$^2$ per year of age (95% CI 0.16–0.23, p<0.001). Almost

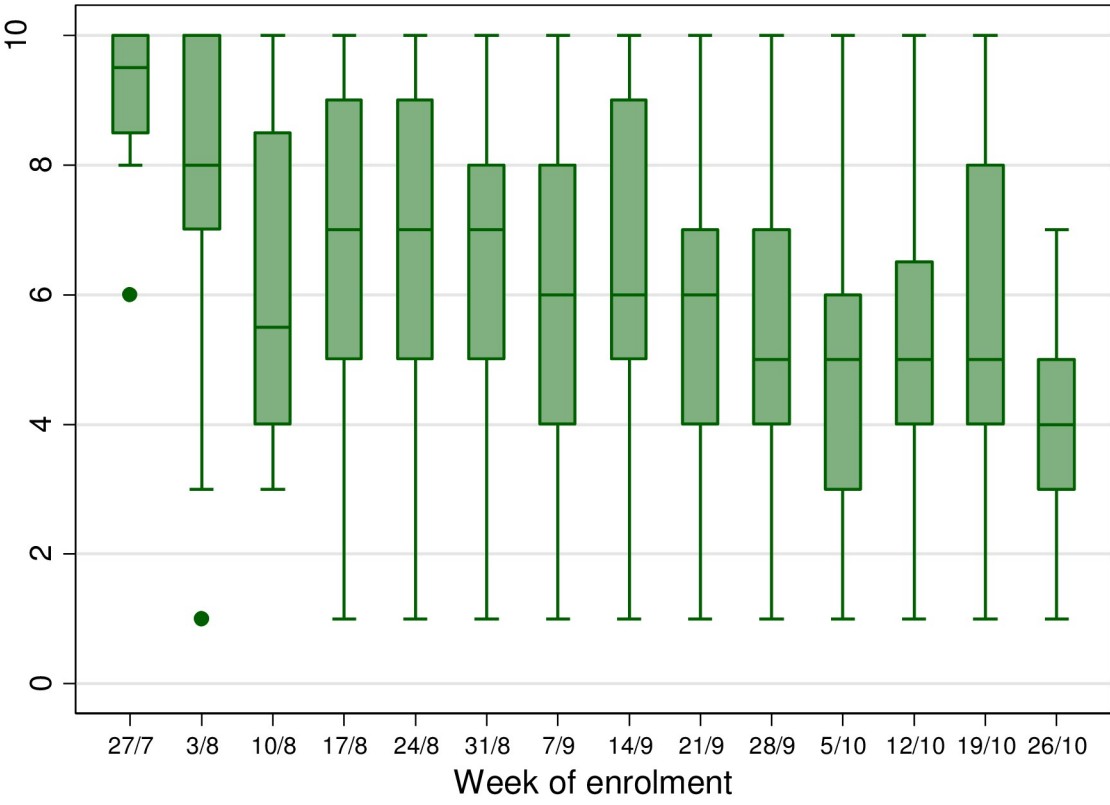

**Fig 2. Healthcare workers' responses to the question "On a scale of 1–10, how serious is the COVID-19 situation" by week of enrolment in the study.** Higher scores indicate higher severity. *(All dates—2020).*

**Table 3. Prevalence of chronic diseases identified by screening health care workers.**

| Medical condition Screening test (number screened) | Number (%) | Of those screened, number (%) with *new* abnormal finding[1] |
|---|---|---|
| Obesity: BMI kg/m$^2$ (n = 948) | | |
| Underweight <18.5 | 24 (2.5) | NA |
| Normal weight 18.5 to <25 | 342 (36.1) | |
| Overweight 25 to <30 | 319 (33.6) | |
| Obese 30+ | 263 (27.7) | |
| Hypertension: BP mmHg (n = 949) | | |
| Normal BP <140/90 | 606 (63.9) | NA |
| Raised BP ≥140/90 | 343 (36.1) | 260/949 (27.4%) |
| Diabetes: HbA1c % (n = 879) | | |
| Normal <6.0% | 607 (69.1) | NA |
| Pre-diabetes 6.0–6.4% | 184 (20.9) | 181/881 (20.5%) |
| Diabetes ≥6.5% | 88 (10.0) | 79/879 (9.0%) |
| Common mental disorder: Shona Symptom Questionnaire-14 (SSQ-14) (n = 951) | | |
| SSQ <8 | 881 (92.6) | NA |
| SSQ ≥8 | 70 (7.4) | 70/951 (7.4%) |
| HIV: Onsite HIV blood test (n = 458)[2] | | |
| Negative | 456 (99.6) | NA |
| Positive | 2 (0.4) | 2/458 (0.4%) |
| TB: GeneXpert MTB/RIF (n = 9) | | |
| Negative | 9 (100.0) | NA |

CI: Confidence interval. BMI: Body Mass Index. BP: blood pressure. TB: tuberculosis.

[1] Excluding those with previous diagnosis.

[2] A further 76 oral self-tests were performed, for which results are not available.

two thirds (517/781, 66%) of female clients had a BMI of 25 or above, compared to 65/167 (39%) of men (odds ratio [OR] adjusted for age 3.5, 95% CI 2.4–5.1, p<0.001).

At enrolment, 343 (36%) had a blood pressure of 140/90mmHg or above, of whom 83 (24%) had a pre-existing diagnosis of hypertension. Eighteen clients (1.9%) had severe hypertension, defined as systolic blood pressure of 180mmHg or above, and/or diastolic blood pressure of 120mmHg or above. Independent risk factors for raised blood pressure included age (adjusted OR [aOR] 1.05 per year of age, 95% CI 1.03–1.07, p<0.001), male sex (aOR 1.89, 95% CI 1.31–2.74, p = 0.001) and having a raised BMI (aOR 2.00, 95% CI 1.46–2.74, p<0.001). In the cohort, 121 had a pre-existing diagnosis of hypertension, meaning that 83/121 (69%) of those with *known* hypertension had a raised blood pressure at screening, including 76/112 (68%) of those who reported taking antihypertensive medications. Adding those with a previous diagnosis of hypertension but normal blood pressure at enrolment, the overall estimated prevalence of hypertension in healthcare workers was 40% (95% CI 37–43%).

Of 881 clients who had an HbA1c measured, 184 (21%) had pre-diabetes (HbA1c 6.0–6.4%), and 88 (10%) had diabetes (HbA1c ≥6.5%). Of these 88, only 9 (10%) had been previously diagnosed with diabetes. The odds of having an HbA1c ≥6.5% increased with age (aOR 1.04 per year of age, 95% CI 1.02–1.07, p = -0.001), but was not significantly associated with male sex (aOR 0.58, 95% CI 0.29–1.12, 0.13) or of having a BMI above 25 (OR 1.16, 95% CI 0.70–1.92, p = 0.57). Of the 17 clients in the cohort who reported a previous diagnosis of diabetes, nine (53%) had an HbA1c≥6.5%, including 6 (46%) of those taking oral antidiabetic

medications, and 3/5 (60%) of those taking insulin. Including those with a known diagnosis of diabetes, the estimated prevalence of diabetes in healthcare workers was 10% (95% CI 8–12%).

In total, 70 clients (prevalence 7%, 95% CI 6–9%) had an SSQ-14 score ≥8, suggestive of a common mental disorder: 8% of female compared to 2% of male clients (aOR 3.75, 95% CI 1.35–10.4, p = 0.011). Nine (0.9%) clients had a very high score of 11 or above. No client reported a pre-existing diagnosis of mental health problems. There was no association with age (aOR 1.01 per year of age, 95% CI 0.98–1.04) or level of concern about COVID-19 (aOR 0.98, 95% CI 0.98–1.08). Of those with an SSQ-14 score ≥ 8, 44% reported being "fearful" or "very fearful" of COVID-19, compared to 34% of those with an SSQ-14 score <8 (p = 0.24).

Of 881 clients without known HIV, 457 (52%) received a provider-performed blood-based HIV test. Of these, two clients tested positive for HIV (0.4%). A further 76 (9%) clients not known to have HIV accepted an oral HIV test for self-testing off-site (the results of which are unknown); the remaining 348 (39%) did not want an HIV test. Including those who were known to be HIV-positive prior to enrolment, the HIV prevalence in healthcare workers was 8% (95% CI 6–9%).

Overall 7% (69/951) of clients reported at least one TB symptom but most were unable to produce a sputum sample. Of nine sputum samples tested using Xpert MTB/Rif, all were negative.

## Discussion

Healthcare worker incidence of SARS-CoV2 infection mirrored the national incidence over the period the services were offered. Worldwide healthcare workers are known to be at high risk of SARS-CoV-2 infection [1–3]. However, in Zimbabwe a high proportion of healthcare workers had only limited patient contact during the study period, due to the hospital operating in emergency mode and ongoing staff strikes, which limited inpatient activity. It is therefore likely that the SARS-CoV-2 infections identified by this study reflect transmission within the community.

Despite good knowledge about SARS-CoV-2 transmission and prevention, healthcare workers reported significant challenges in protecting themselves against the virus at work and at home. The majority of clients stated that running water was not always available in their workplace, and while most of them had been given a mask at some point, they were not always available. A 2019 WHO/UNICEF report found that 19% of healthcare facilities in Zimbabwe (and 26% globally) lack basic water services, and those that exist may be intermittent or unreliable [11]. This poses a significant challenge to IPC, both for SARS-CoV-2 and for other infectious pathogens, and highlights an urgent need for improved infrastructure [26]. It was clear that clients' ability to protect themselves was also impacted by circumstances outside the healthcare system, with low wages, increasing cost-of-living and lack of health information cited as important challenges. The SARS-CoV-2 pandemic and associated control measures are predicted to cause significant economic damage across the continent [27,28] with potentially profound socioeconomic implications [8,29]; healthcare workers are not immune from the wider socioeconomic forces impacting their communities.

SARS-CoV-2 testing was primarily offered to those with symptoms or fever, although in practice a small number of clients without symptoms were also tested (particularly clients who reported significant contact with someone with SARS-CoV-2). The true rates of SARS-CoV-2 infection may therefore have been higher. Asymptomatic and pauci-symptomatic infection is well-recognised, and transmission by people without symptoms may account for around half of SARS-CoV-2 transmission [30]. Notably a third of the healthcare workers testing positive for SARS-CoV-2 in this study had no symptoms. In the healthcare context a high rate of

minimally symptomatic infections makes it more challenging to prevent transmission between healthcare workers and potentially to vulnerable patients, particularly in the context of suboptimal PPE [3]. Notably 29% of healthcare workers in this study did not know that the virus could be transmitted by people without symptoms.

Reassuringly all clients with a positive SARS-CoV-2 test had mild disease; none required hospitalisation. The young age of this cohort may have been protective against severe disease; however, there was an appreciable prevalence of potential risk factors for severe COVID-19. For example, the majority of healthcare workers were overweight or obese, a third had elevated blood pressure readings and 10% had diabetes. This is worrying because diabetes, hypertension and obesity have all been associated with increased risk of severe COVID-19 [19–21].

COVID-19 threatens the psychological as well as physical wellbeing of healthcare workers globally [31], and studies have reported extremely high prevalence of psychological symptoms in frontline healthcare workers [6]. In this study, healthcare workers reported COVID-19 as a serious concern, but the perceived severity declined over time, again mirroring the trend in case numbers nationally. The proportion of workers with a raised SSQ-14, suggestive of common mental disorder, was stable over time and not significantly associated with their feelings about COVID-19. This may reflect a high pre-existing burden of mental illness in general: healthcare workers in Zimbabwe are subject to continuous pressures from a precarious socioeconomic context and challenging working environments, independent of the current pandemic.

This study demonstrates the feasibility of conducting comprehensive occupational health screening during a pandemic. Access to occupational health services is often limited for healthcare workers in low- and middle- income countries, despite them being at high risk of infectious diseases and critical for the function of the wider healthcare system. Identification and treatment of chronic conditions in healthcare workers demands additional attention currently because these conditions are associated with severe COVID-19 [19–21]. In this study, healthcare workers had a similar prevalence of hypertension to the general population in Zimbabwe [16], while the 10% diabetes prevalence was higher than a previous estimate of the Zimbabwean pooled population prevalence of 5.7% [17]. Most (e.g. 75% of those with high blood pressure, 90% of those with diabetes) were undiagnosed, suggesting that screening healthcare workers for non-communicable diseases and risk factors may facilitate earlier diagnosis. However, this must go hand in hand with provision of effective treatment and follow-up. Half of those with known diabetes had an HbA1c above 6.5%, and two-thirds of those reporting known hypertension had a raised blood pressure at screening, including many individuals who were on treatment; this suggests that control of long-term conditions is inadequate even when diagnosed.

Unlike other comorbidities, coverage of prior HIV screening was high, with 96% of clients reporting a previous HIV test. There was a high acceptability of HIV testing within the occupational health service, with more than half of clients without an HIV diagnosis accepting a blood-based test, and a further 9% accepting an oral self-test. The overall HIV prevalence of 8% was slightly lower than Zimbabwe's adult HIV prevalence of 12.7% [15]. Offering HIV screening to healthcare workers did identify additional cases. Nevertheless, overall 97% of HIV infections were previously diagnosed, and all with known HIV reported current ART. This reflects robust programming and investment in HIV testing and treatment services, and contrasts with the high burden of undiagnosed non-communicable disease seen in this cohort. Globally, non-communicable diseases account for 60% of disability-adjusted life-years lost but receive less than 2% of overseas development assistance for health (compared to 3% and 30% respectively for HIV) [32].

There was a 7% prevalence of TB symptoms, but rates of TB testing were low, largely because clients were unable to produce sputum. Providers may also have been prioritising SARS-CoV-2 testing for those with respiratory symptoms.

This study has several important strengths. A high number of healthcare workers were screened over a very short period of time, appropriate screening tools were used, and all testing was done by trained research staff. This is the first report of a large comprehensive occupational health screening programme in Zimbabwe, and one of few reports covering SARS-CoV-2 from the region, adding to our understanding of the health and wellbeing of healthcare workers in the pandemic.

There were several important limitations. There was significant selection bias in those self-referring to the service; particularly early in the study, symptomatic clients who were concerned about SARS-CoV-2 were over-represented. Due to limitations in testing capacity nationally, testing was targeted to those with symptoms; some asymptomatic SARS-CoV-2 infections may have been missed. Thirdly, recruitment coincided with the first epidemic peak in Zimbabwe, and began four months after the first cases were reported in the country. It is likely that some clients may have had infection before the service was initiated; results from other settings have shown healthcare workers may be infected early in epidemics [33]. Serological testing may help to clarify the true burden of SARS-CoV-2 infection in Zimbabwean healthcare workers. Thirdly, occupational health assessment was necessarily limited to screening; a single blood-pressure reading is not sufficient to diagnose hypertension, nor a raised SSQ-14 score to diagnose mental illness. However, they are appropriate screening tests (SSQ-14 was developed and validated for use in Zimbabwe [24]) to prompt referral for more thorough assessment. Clients were referred for further care but data on linkage to care were not available, which limits our ability to assess effectiveness of screening, particularly as control of hypertension and diabetes was poor in a high proportion of clients previously diagnosed.

## Conclusion

We observed important levels of SARS-CoV-2 infection amongst Zimbabwean healthcare workers, which declined in line with community incidence. All infections were non-severe, and many were pauci- or asymptomatic. Healthcare workers highlighted the lack of resources for IPC and wider socioeconomic challenges as important barriers to protecting themselves from SARS-CoV-2. While healthcare workers were concerned about COVID-19, their perception of the seriousness of the pandemic waned over time.

It was feasible to offer comprehensive occupational health services during a pandemic to a large number of healthcare workers. SARS-CoV-2 testing offered an opportunity to integrate comprehensive screening for other common conditions, revealing a considerable burden of previously undiagnosed comorbidities which may increase the risk of severe COVID-19 and cause morbidity in their own right. Despite themselves being healthcare workers, a third of clients lacked medical insurance, thereby limiting their own access to care. Healthcare workers are critical in the global fight against COVID-19, and it is essential to support their physical and psychological wellbeing during the pandemic.

## Supporting information

**S1 Table. Demographic and clinical characteristics of participants not tested, testing negative, and testing positive for, SARS-CoV-2.**
(DOCX)

**S1 File. Anonymised datafile for participants.** Certain demographic indicators have been removed or categorised to preserve participant confidentiality.
(XLSX)

**S2 File. Copy of electronic case report forms used in data collection.**
(PDF)

## Author Contributions

**Conceptualization:** Grace McHugh, Chiratidzo Ndhlovu, Hilda Mujuru, Simbarashe Rusakaniko, Portia Manangazira, Rashida A. Ferrand, Katharina Kranzer.

**Data curation:** Hannah M. Rickman, Rudo Chingono, Tinotenda Taruvinga, Tsitsi Bandason, Nicol Redzo, Katharina Kranzer.

**Formal analysis:** Hannah M. Rickman, Rudo Chingono, Tinotenda Taruvinga, Tsitsi Bandason, Nicol Redzo, Katharina Kranzer.

**Funding acquisition:** Celia L. Gregson, Rashida A. Ferrand.

**Investigation:** Fungai Kavenga, Rudo Chingono, Tinotenda Taruvinga, Takudzwa Marembo, Justen Manasa, Tsitsi Magure.

**Methodology:** Rudo Chingono, Tinotenda Taruvinga, Takudzwa Marembo, Justen Manasa, Grace McHugh, Tsitsi Bandason, Nicol Redzo, Aspect Maunganidze, Tsitsi Magure, Rashida A. Ferrand.

**Project administration:** Fungai Kavenga, Rudo Chingono, Tinotenda Taruvinga, Edson Marambire, Nicol Redzo, Katharina Kranzer.

**Resources:** Takudzwa Marembo, Aspect Maunganidze, Tsitsi Magure, Katharina Kranzer.

**Supervision:** Fungai Kavenga, Rudo Chingono, Katharina Kranzer.

**Visualization:** Hannah M. Rickman.

**Writing – original draft:** Hannah M. Rickman, Katharina Kranzer.

**Writing – review & editing:** Fungai Kavenga, Hannah M. Rickman, Rudo Chingono, Tinotenda Taruvinga, Takudzwa Marembo, Justen Manasa, Edson Marambire, Grace McHugh, Celia L. Gregson, Tsitsi Bandason, Nicol Redzo, Aspect Maunganidze, Tsitsi Magure, Chiratidzo Ndhlovu, Hilda Mujuru, Simbarashe Rusakaniko, Portia Manangazira, Rashida A. Ferrand, Katharina Kranzer.

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
