## [Decision Letter · Decision Letter 0]

29 Mar 2021

PONE-D-21-03459

Comprehensive occupational health services for front-line healthcare workers in Zimbabwe during the SARS-CoV-2 pandemic

PLOS ONE

Dear Dr. Hannah Rickman,

Thank you for submitting your manuscript to PLOS ONE. After careful consideration, we feel that it has merit but does not fully meet PLOS ONE’s publication criteria as it currently stands. Therefore, we invite you to submit a revised version of the manuscript that addresses the points raised during the review process.

We look forward to receiving your revised manuscript.

Kind regards,

Claudia Marotta

Academic Editor

PLOS ONE

Journal Requirements:

3. Please provide additional details regarding participant consent. In the ethics statement in the Methods and online submission information please specify how verbal/oral consent was recorded. If your study included minors, please state whether you obtained consent from parents or guardians in these cases. If the need for consent was waived by the ethics committee, please include this information.

Futhermore, please include a copy of the baseline questionnaire used as a part of the study as Supporting File.

Additional Editor Comments (if provided):

dear authors follow reviewer suggestions to improve your paper

Reviewers' comments:

Reviewer's Responses to Questions

**Comments to the Author**

1. Is the manuscript technically sound, and do the data support the conclusions?

Reviewer #1: Yes

Reviewer #2: Partly

2. Has the statistical analysis been performed appropriately and rigorously? 

Reviewer #1: I Don't Know

Reviewer #2: Yes

3. Have the authors made all data underlying the findings in their manuscript fully available?

Reviewer #1: No

Reviewer #2: Yes

4. Is the manuscript presented in an intelligible fashion and written in standard English?

Reviewer #1: Yes

Reviewer #2: Yes

5. Review Comments to the Author

Reviewer #1: It looks good sound as first National study, you already mentioned your limitations, be sure about reviewing the language and pilgiarism. Also you can add to your recommendations part related to National policy for health care workers.

Reviewer #2: Critical review of the manuscript titled

”Comprehensive occupational health services for front-line healthcare workers in Zimbabwe during the SARS-CoV-2 pandemic”

submitted to Archives of Environmental & Occupational Health

The authors have set up an occupational health screening service for healthcare workers (HCWs) in Harare, Zimbabwe, in the summer of 2020 at the peak of the first wave of COVID-19 pandemic in the country. The service was voluntarily visited by HCWs from Harare healthcare facilities, where they received a complex screening including symptoms and testing of a series of communicable (SARS-CoV-2, HIV, TB) and non-communicable (obesity, hypertension, diabetes, mental disorders) diseases. The effort is appreciable and the findings in the 27th July and 30th October period are well analysed and presented. The manuscript is written in perfect scientific style and format.

I have only a few major and minor comments related to limitations and possible additional analysis of study findings.

Major comments

1) The authors mention that the voluntary participation in the screening could introduce selection bias, which certainly exist. To that an important additional information is the size of the source/target population, which is not reported. How big is the source population, i.e. healthcare staff in Harare and at the Parirenyatwa Group of Hospitals?

2) How was the occupational screening service advised? Did the advertisement reach the target population evenly?

3) The title names front-line HCWs as the study population; however, the screening was offered free-of-charge for any HCWs, if I understand correctly. How was the participation of front-line HCWs assured?

4) “All clients were asked about symptoms of COVID-19 including fever, cough, coryzal symptoms, joint pain, headache and loss of smell or taste.” (page6, lines 19-20). How was the set of symptoms determined? E.g. fatigue and chest tightness are also typical symptoms of COVID-19 disease, but not on the list.

5) Who stayed out of SARS-CoV-2 testing? Were they different in any characteristics from those, who accepted testing?

6) “A further 23 clients without symptoms were also tested, of whom 20 (87%) reported contact with someone with COVID-19.” (page10, lines 18-19). It would be interesting to see how the proportion of positive PCR test results varied between groups of different indications for testing.

7) On Figure 1, the proportion of positive results from those tested would give a more valid picture since the absolute number of attendances to the screening service changed with time.

8) Similarly to the tested non-communicable disease, the potential risk factors of COVID-19 positivity could also be analysed (i.e. availability and use of preventive measures, knowledge and attitude), even though there were only a limited number of cases.

9) Was there a question on the possible occupational exposure to COVID-19 patents in a healthcare setting? The authors speculate that most of the SARS-CoV-2 infections are probably community transmissions. It would be interesting to estimate it from available data.

10) “All infections were non-severe, and many were pauci- or asymptomatic.” (page 17, lines 22-23). The study provides only a snapshot about the status of the positively tested participants. Is there follow-up information about the pauci- and asymptomatic cases? Some of them may have developed more severe symptoms later or became long-COVID cases.

Minor comments

1) “While the clients waited for screening, a trained researcher administered a baseline questionnaire, which included questions on past medical history, knowledge about COVID-19 prevention and management, attitudes and behaviours related to COVID-19 and availability of PPE and other resources in their workplace.” (page 8, lines 2-5). What about the SSQ questionnaire? Was it not administered with the baseline questionnaire?

2) “Overall 7% (69/951) of clients reported at least one TB symptom but most were unable to produce a sputum sample.” (page 13, lines 16-17). What does “unable to produce a sputum sample” mean?

3) “Secondly, recruitment coincided…” (page 17, line 9). It is actually the third limitation.

4) “We observed significant levels of SARS-CoV-2 infection amongst Zimbabwean healthcare workers,…” (page 17, lines 21-22). The significance of the prevalence of infection depends on the base for comparison. The rate is not high compared to many other places and times.

Summary

The manuscript discusses the observations of a very important occupational health service for HCWs during the COVID-19 pandemic. The data are correctly analysed and interpreted, and the manuscript is well-written. However, there is room for some clarification and extension of analysis as suggested in the comments.

6. PLOS authors have the option to publish the peer review history of their article (what does this mean?). If published, this will include your full peer review and any attached files.

Reviewer #1: No

Reviewer #2: No

---

## [Author Response · Author response to Decision Letter 0]

1 Jun 2021

Dear Editor, 

Thank you for considering our manuscript for publication and for sending comments from peer reviewers. Please find below a detailed response to the editorial comments and reviewers’ comments. 

We have double checked the requirements and have made the necessary changes. 

We are committed to providing open access data, within the confines of producing a de-identified minimal dataset. All participants in the study are healthcare workers within a clearly identifiable (and named) healthcare facility, and there is a substantial risk that individuals could be identified because of a combination of demographic characteristics (e.g. sex and occupation). The principal investigator and collaborators felt that there was an unacceptable risk to participant confidentiality, particularly as these data include sensitive health information (including about HIV status and other chronic illnesses). We have therefore produced a deidentified dataset in which information on occupation and sex has been removed, and ages replaced with age categories, permitting the reproduction of almost all analyses without compromising confidentiality. This follows the guidelines at http://www.bmj.com/content/340/bmj.c181.long as suggested. This dataset has been submitted as part of the re-submission. 

3. Please provide additional details regarding participant consent. In the ethics statement in the Methods and online submission information please specify how verbal/oral consent was recorded. If your study included minors, please state whether you obtained consent from parents or guardians in these cases. If the need for consent was waived by the ethics committee, please include this information.

Furthermore, please include a copy of the baseline questionnaire used as a part of the study as Supporting File.

Thank you. All participants were adults aged over 18; this has been added to the methods. Verbal consent was recorded electronically as part of the clinical questionnaire, with the ethics committees waiving a requirement for written consent; this is documented in the methods section. Questionnaire included as a supporting file. 

REVIEWER COMMENTS

Major comments

1) The authors mention that the voluntary participation in the screening could introduce selection bias, which certainly exist. To that an important additional information is the size of the source/target population, which is not reported. How big is the source population, i.e. healthcare staff in Harare and at the Parirenyatwa Group of Hospitals?

Parirenyatwa Group of Hospitals employs an excess of 2000 staff. However, most of the domestic workers and security staff are employed by private companies. In addition, students including those studying nursing, medicine and laboratory sciences, who had access to the screening service are not included in the staff estimates. Furthermore, while Parirenyatwa Group of Hospitals employs more than 2000 staff, the staff contingent actually working during the study period was much smaller. As outlined in the manuscript “As a result of the national lockdown, the facility moved to “emergency operations”, with outpatient services closed from the end of March 2020. Provision of other health services was limited by the industrial action by doctors and nurses.” Given the limitations regarding the size of the catchment population we are unable to provide an estimate about the proportion reached with this service. We have added a line to the methods to explain this. 

2) How was the occupational screening service advised? Did the advertisement reach the target population evenly?

We have added a line in the methods to explain how the service was advertised. All care was taken to ensure that news of the service reached all work groups, but this was a pragmatic study and data were not collected on how advertisement reached different groups.

3) The title names front-line HCWs as the study population; however, the screening was offered free-of-charge for any HCWs, if I understand correctly. How was the participation of front-line HCWs assured?

We appreciate that the distinction of a “front-line” healthcare worker may be misleading, so have removed this descriptor from the title and conclusion. The service was set up to be inclusive from the start, i.e. we encouraged domestics, security staff and also people working as part of the estates (for example those working in the laundry and operating the incinerator) to access the services. These staff members may not be considered “front-line” health care workers, but may have a considerable risk of being exposed to SARS-CoV-2. 

4) “All clients were asked about symptoms of COVID-19 including fever, cough, coryzal symptoms, joint pain, headache and loss of smell or taste.” (page6, lines 19-20). How was the set of symptoms determined? E.g. fatigue and chest tightness are also typical symptoms of COVID-19 disease, but not on the list.

Thank you – a fuller list of symptoms was also included but is not listed in full in the manuscript – please see questionnaire in Supplementary Materials for more information. At the time of recruitment there was not an internationally-accepted list of symptoms suggestive of possible COVID, so this list was derived from available guidance and consensus medical opinion. 

5) Who stayed out of SARS-CoV-2 testing? Were they different in any characteristics from those, who accepted testing?

Few participants declined SARS-CoV-2 testing when offered (particularly as there was very limited availability of SARS-CoV-2 testing in Zimbabwe at the time), but data on this were not specifically collected. Of the 66 clients with COVID-19 symptoms, six were not tested. We have included the descriptive characteristics of those not tested for SARS-CoV-2, those testing negative, and those testing positive, as a supplementary table (Supplementary Table 1). 

6) “A further 23 clients without symptoms were also tested, of whom 20 (87%) reported contact with someone with COVID-19.” (page10, lines 18-19). It would be interesting to see how the proportion of positive PCR test results varied between groups of different indications for testing.

We have included the descriptive characteristics (including contact history) of those not tested for SARS-CoV-2, those testing negative, and those testing positive, as a supplementary table.

7) On Figure 1, the proportion of positive results from those tested would give a more valid picture since the absolute number of attendances to the screening service changed with time.

Thank you for the suggestion. We wanted also to demonstrate the finding that, during the first weeks of the service, symptomatic clients predominated leading to a high proportion being tested, and that the balance of service users changed over time as well as the proportion testing positive, and therefore chose to present the data as in Figure 1. 

8) Similarly to the tested non-communicable disease, the potential risk factors of COVID-19 positivity could also be analysed (i.e. availability and use of preventive measures, knowledge and attitude), even though there were only a limited number of cases.

We considered this, but the low number of cases precluded any meaningful statistical analysis and we therefore did not include this within the main body of the analysis. As the occupational health service continues to operate and to identify additional cases, this may be possible in future analyses. We have included the descriptive characteristics of those not tested for SARS-CoV-2, those testing negative, and those testing positive, as a supplementary table. 

9) Was there a question on the possible occupational exposure to COVID-19 patents in a healthcare setting? The authors speculate that most of the SARS-CoV-2 infections are probably community transmissions. It would be interesting to estimate it from available data.

We agree that this is an interesting question. Our questionnaire only asked about contact to those with COVID-19, and did not differentiate between occupational and non-occupational exposure. This may be changed for future iterations. 

10) “All infections were non-severe, and many were pauci- or asymptomatic.” (page 17, lines 22-23). The study provides only a snapshot about the status of the positively tested participants. Is there follow-up information about the pauci- and asymptomatic cases? Some of them may have developed more severe symptoms later or became long-COVID cases.

All participants were followed up until symptoms had fully resolved, and until 14 days from their positive test. We have expanded in the results. Of course it is possible that some participants may have subsequently developed post-COVID syndromes, but at the time follow-up ended all participants were well.

Minor comments

1) “While the clients waited for screening, a trained researcher administered a baseline questionnaire, which included questions on past medical history, knowledge about COVID-19 prevention and management, attitudes and behaviours related to COVID-19 and availability of PPE and other resources in their workplace.” (page 8, lines 2-5). What about the SSQ questionnaire? Was it not administered with the baseline questionnaire?

Thank you – we have clarified in the methods section. Clients were asked to fill the SSQ questionnaire while they were waiting. The questionnaire was self-administered to ensure that clients had enough time to consider the questions and minimise social-desirability bias. The SSQ questionnaire includes questions which some people may not feel comfortable to answer truthfully when asked by another person, for examples questions on self-harm and crying. The client submitted to questionnaire to the nurse – who checked the score and looked for any answers which would raise red flags. 

2) “Overall 7% (69/951) of clients reported at least one TB symptom but most were unable to produce a sputum sample.” (page 13, lines 16-17). What does “unable to produce a sputum sample” mean?

Even amongst symptomatic, hospitalised or immunosuppressed cohorts with presumptive TB, a high proportion are unable to spontaneously expectorate a sputum sample. One might expect this proportion to be even higher amongst minimally-symptomatic clients identified in untargeted screening. For infection prevention and control purposes a decision was taken not to perform sputum induction. 

3) “Secondly, recruitment coincided…” (page 17, line 9). It is actually the third limitation.

Thank you; amended. 

4) “We observed significant levels of SARS-CoV-2 infection amongst Zimbabwean healthcare workers,…” (page 17, lines 21-22). The significance of the prevalence of infection depends on the base for comparison. The rate is not high compared to many other places and times.

We have edited this to “important” to avoid the suggestion of a statistical comparison.

---

## [Decision Letter · Decision Letter 1]

15 Jul 2021

PONE-D-21-03459R1

Comprehensive occupational health services for healthcare workers in Zimbabwe during the SARS-CoV-2 pandemic

PLOS ONE

Dear Dr. Rickman,

Thank you for submitting your manuscript to PLOS ONE. After careful consideration, we feel that it has merit but does not fully meet PLOS ONE’s publication criteria as it currently stands. Therefore, we invite you to submit a revised version of the manuscript that addresses the points raised during the review process.

ACADEMIC EDITOR: Please review comments made by the reviewer and provide your response in the revised manuscript.

We look forward to receiving your revised manuscript.

Kind regards,

Muhammad Adrish, MD, MBA, FCCP, FCCM

Academic Editor

PLOS ONE

Journal Requirements:

Reviewers' comments:

Reviewer's Responses to Questions

**Comments to the Author**

1. If the authors have adequately addressed your comments raised in a previous round of review and you feel that this manuscript is now acceptable for publication, you may indicate that here to bypass the “Comments to the Author” section, enter your conflict of interest statement in the “Confidential to Editor” section, and submit your "Accept" recommendation.

Reviewer #2: (No Response)

2. Is the manuscript technically sound, and do the data support the conclusions?

Reviewer #2: Yes

3. Has the statistical analysis been performed appropriately and rigorously? 

Reviewer #2: Yes

4. Have the authors made all data underlying the findings in their manuscript fully available?

Reviewer #2: No

5. Is the manuscript presented in an intelligible fashion and written in standard English?

Reviewer #2: Yes

6. Review Comments to the Author

Reviewer #2: Critical review of the revised manuscript titled

”Comprehensive occupational health services for healthcare workers in Zimbabwe during the SARS-CoV-2 pandemic”

submitted to PLOS ONE

The authors have responded to all the major and minor comments in an acceptable way. Nevertheless, the place of the modifications made in the manuscript are not specified in the responses (line numbers), neither highlighted in the text of the manuscript (track changes or coloured background). Without this, nowadays routinely provided, guiding information I am not able to follow the amendments of the manuscript. I could also find access to the supplementary materials, too, that have also been modified during the revision.

I would like to request this additional information from the authors to be able to make a final judgement on the manuscript.

7. PLOS authors have the option to publish the peer review history of their article (what does this mean?). If published, this will include your full peer review and any attached files.

Reviewer #2: No

---

## [Author Response · Author response to Decision Letter 1]

6 Oct 2021

Dear Editor, 

Thank you for considering our manuscript for publication and apologies for the delayed response to comments.

Please see below an amended version of the initial response to reviewers, which now includes line references for all changes. As requested by PLOS One, we had uploaded both a tracked changes and a clean, unmarked version of the manuscript, which were both included in the previous submission. The line references refer to the tracked changes version (included second in the submission PDF). We hope this will make the response easier to follow, but please do contact us if there are any additional concerns.

We are sorry to hear there were problems with the supplementary materials – these had been uploaded with the previous submission, but we have contacted the journal to make sure these are all accessible and there are no other issues.

Kind regards,

Dr Hannah Rickman

---

## [Decision Letter · Decision Letter 2]

8 Nov 2021

Comprehensive occupational health services for healthcare workers in Zimbabwe during the SARS-CoV-2 pandemic

PONE-D-21-03459R2

Dear Dr. Rickman,

We’re pleased to inform you that your manuscript has been judged scientifically suitable for publication and will be formally accepted for publication once it meets all outstanding technical requirements.

Kind regards,

Akihiro Nishi, M.D., Dr.P.H.

Academic Editor

PLOS ONE

Additional Editor Comments (optional):

The two reviewers upon the original and second submissions are satisfied. I am happy to suggest Accept at this moment.

Reviewers' comments:

Reviewer's Responses to Questions

**Comments to the Author**

1. If the authors have adequately addressed your comments raised in a previous round of review and you feel that this manuscript is now acceptable for publication, you may indicate that here to bypass the “Comments to the Author” section, enter your conflict of interest statement in the “Confidential to Editor” section, and submit your "Accept" recommendation.

Reviewer #2: All comments have been addressed

2. Is the manuscript technically sound, and do the data support the conclusions?

Reviewer #2: Yes

3. Has the statistical analysis been performed appropriately and rigorously? 

Reviewer #2: Yes

4. Have the authors made all data underlying the findings in their manuscript fully available?

Reviewer #2: Yes

5. Is the manuscript presented in an intelligible fashion and written in standard English?

Reviewer #2: Yes

6. Review Comments to the Author

Reviewer #2: Critical review of the revised manuscript titled

”Comprehensive occupational health services for healthcare workers in Zimbabwe during the SARS-CoV-2 pandemic”

submitted to PLOS ONE

The authors have provided responses to all the major and minor comments and amended the manuscript at appropriate places. The manuscript is acceptable in the present form. Nevertheless, the authors might consider two further recommendations when finalizing the publication.

• The “Contact of TB case” option is duplicated in Q13 of the questionnaire.

• The answer given to major comment 7) about Figure 1 could be included in the text for clarification.

7. PLOS authors have the option to publish the peer review history of their article (what does this mean?). If published, this will include your full peer review and any attached files.

Reviewer #2: No

---

## [Editor Report · Acceptance letter]

15 Nov 2021

PONE-D-21-03459R2 

Comprehensive occupational health services for healthcare workers in Zimbabwe during the SARS-CoV-2 pandemic 

Dear Dr. Rickman:

I'm pleased to inform you that your manuscript has been deemed suitable for publication in PLOS ONE. Congratulations! Your manuscript is now with our production department. 

Kind regards, 

on behalf of

Dr. Akihiro Nishi 

Academic Editor

PLOS ONE